# RECOVERY OF TRAINING DATA FROM OVERPARAMETERIZED AUTOENCODERS: AN INVERSE PROBLEM PERSPECTIVE

## ABSTRACT

We study the recovery of training data from overparameterized autoencoder models. Given a degraded training sample, we define the recovery of the original sample as an inverse problem and formulate it as an optimization task. In our inverse problem, we use the trained autoencoder to implicitly define a regularizer for the particular training dataset that we aim to retrieve from. We develop the intricate optimization task into a practical method that iteratively applies the trained autoencoder and relatively simple computations that estimate and address the unknown degradation operator. We evaluate our method for blind inpainting where the goal is to recover training images from degradation of many missing pixels in an unknown pattern. We examine various deep autoencoder architectures, such as fully connected and U-Net (with various nonlinearities and at diverse train loss values), and show that our method significantly outperforms previous methods for training data recovery from autoencoders. Importantly, our method greatly improves the recovery performance also in settings that were previously considered highly challenging, and even impractical, for such retrieval.

## 1 INTRODUCTION

Deep neural networks (DNNs) are usually overparameterized models with many trainable parameters that can lead to memorization of their training samples. This modern overparameterized regime differs from the classical regime where overfitting is avoided (by small models, strong regularization, etc.) and the general data pattern is learned without significant memorization. Consequently, training data memorization raises potential privacy issues, including the extraction of private training data from the trained DNN.

Recent research works showcased the ability to extract training data from generative DNNs. Carlini et al. (2021) showed that generative language models like GPT-2 can unintentionally memorize and, later, accurately generate sentences from their training data without being explicitly programmed to do so. This indicates that private text data used to train large language models could be extracted or reconstructed by analyzing the trained model. For images, Carlini et al. (2023) showed that generative diffusion models can memorize individual images from their training data and emit them at generation time.

Another line of works (Radhakrishnan et al., 2018a;b; 2020; Jiang & Pehlevan, 2020; Nouri & Seyyedsalehi, 2023) utilizes the associative memory of overparameterized autoencoder DNNs to recover their training data from degraded forms. The autoencoder architecture typically consists of an encoder that maps the input to a latent space representation of a lower dimension, and a decoder that reconstructs the input from the latent representation back to the input space. The training loss often seeks to minimize the mean squared error (MSE) between the training inputs and their reconstructed outputs. This architecture and training procedure can lead to associative memorization of training data as attractors, i.e., fixed points (of the autoencoder function) that inputs from their surrounding region can converge to by iterative application of the autoencoder. Accordingly, Radhakrishnan et al. (2018a;b; 2020); Jiang & Pehlevan (2020); Nouri & Seyyedsalehi (2023) showed that a training sample can be recovered from its degarded version by a simple iterative application of the trained autoencoder. This approach to training data recovery was proved useful only under restrictive

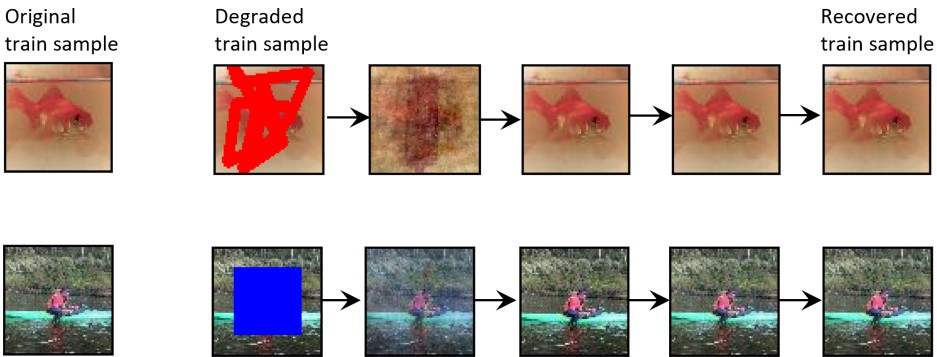

Figure 1: Iterative reconstruction of degraded training images using our proposed approach

conditions such as very specific activation functions, sufficiently small training dataset size, and sufficiently low train loss.

In this work, we provide a new perspective on the recovery of training data from overparameterized autoencoders. We define an inverse problem that aims to recover a training sample from its degraded version, while not knowing the degradation operator. This inverse problem is formulated as an optimization with a regularizer that implicitly depends on the overfitting of training data in the trained autoencoder. To practically solve this optimization problem, we employ the alternating direction method of multipliers (ADMM) (Boyd et al., 2011) and extend the plug-and-play priors (Venkatakrishnan et al., 2013) to our task. Whereas the plug-and-play priors concept is usually used for solving imaging inverse problems based on iterative application of a black-box image denoiser (Venkatakrishnan et al., 2013; Sreehari et al., 2016; Dar et al., 2016; Rond et al., 2016), we use it to solve our training data recovery problem based on the black-box application of the given autoencoder (which is not necessarily trained to be a denoiser). Consequently, we get an iterative algorithm that not only iteratively applies the trained autoencoder (as in previous works), but also estimates the degradation operator and attempts to invert it.

Our new approach to recovery of autoencoders' training data significantly outperforms previous methods. The recovery rate, i.e., the percentage of the training dataset that can be accurately recovered, is significantly increased by using our method. Moreover, our method is able to recover training data in settings where previous methods performed very weakly and even completely failed to recover training data at all.

While our method succeeds to recover training data from autoencoders at higher training loss values, our results also demonstrate the reduction in the recovery ability as the autoencoder is trained to a higher train loss and less overfits its dataset. This, as well as our other results, are useful to understand the privacy risk of training data recovery in autoencoders.

## 2 RECOVERY OF TRAINING DATA AS AN INVERSE PROBLEM

### 2.1 OVERPARAMETERIZED AUTOENCODERS: A DEFINITION

Let us start with a general definition of an autoencoder.

**Definition 1.** *An autoencoder, $f : \mathbb{R}^d \to \mathbb{R}^d$ can be formulated as*

$$f(\mathbf{x}) = f_{\mathrm{dec}}(f_{\mathrm{enc}}(\mathbf{x})) \ \ for \ \mathbf{x} \in \mathbb{R}^d \tag{1}$$

*where $f_{\mathrm{enc}} : \mathbb{R}^d \to \mathbb{R}^m$ is an encoder from the $d$-dimensional input space to an $m$-dimensional latent space, and $f_{\mathrm{dec}} : \mathbb{R}^m \to \mathbb{R}^d$ is a decoder from the latent space back to the input space.*

We consider an autoencoder $f$ to be overparameterized if it has the ability to perfectly fit its $n$ training samples $\mathbf{x}_1, \ldots, \mathbf{x}_n \in \mathbb{R}^d$. Namely, the perfect fitting condition implies that each of the training samples is a fixed point of the trained autoencoder

$$f(\mathbf{x}_i) = \mathbf{x}_i, \ \ i = 1, \ldots, n. \tag{2}$$

In practice, the perfect fitting condition (2) can be approximate due to numerical precision or early stopping of the training. Moreover, perfect fitting usually requires a latent space dimension (denoted as $m$ in Definition 1) larger than the number of training samples; this condition appeared, for example, in (Radhakrishnan et al., 2020) for autoencoder neural networks, and in (Dar et al., 2020) for PCA-based linear autoencoders.

## 2.2 THE RECOVERY PROBLEM

We are given an overparameterized autoencoder $f$ that was trained to perfectly (or approximately) fit its training data. We also get a degraded version of the autoencoder's $j^{\text{th}}$ training sample

$$\mathbf{y}_j = \mathbf{H}\mathbf{x}_j + \boldsymbol{\epsilon} \tag{3}$$

where $\mathbf{H} \in \mathbb{R}^{d \times d}$ is a deterministic degradation operator that may, e.g., erase components by zeroing the respective entries of a $d$-dimensional vector; and $\boldsymbol{\epsilon} \sim \mathcal{N}(\mathbf{0}, \sigma_\epsilon^2 \mathbf{I}_d)$ is a $d$-dimensional Gaussian noise vector.

The recovery task aims to estimate the original training sample $\mathbf{x}_j$ given its degraded version $\mathbf{y}_j$ and the trained autoencoder $f$. The recovery performance is evaluated in terms of the mean squared error (MSE)

$$\mathcal{E}_{\text{rec}}(\widehat{\mathbf{x}}_j, \mathbf{x}_j) = \frac{1}{d} \|\widehat{\mathbf{x}}_j - \mathbf{x}_j\|_2^2 \tag{4}$$

where $\widehat{\mathbf{x}}_j$ is the estimate of $\mathbf{x}_j$. Note that we consider here the recovery of a single training sample by using only its degraded version and the trained autoencoder. Importantly, the specific degradation operator $\mathbf{H}$ is unknown for the recovery task, although the general type of $\mathbf{H}$ is assumed to be known (e.g., it is known that $\mathbf{H}$ is a pixel erasure operator but without knowing how many and which pixels were erased and what are their new values).

## 2.3 THE INVERSE PROBLEM PERSPECTIVE TO TRAINING DATA RECOVERY

The recovery problem in Section 2.2 can be considered as an inverse problem whose optimization form is

$$\{\widehat{\mathbf{x}}, \widehat{\mathbf{H}}\} = \underset{\mathbf{x} \in \mathbb{R}^d, \widetilde{\mathbf{H}} \in \mathbb{R}^{d \times d}}{\arg\min} \left\|\widetilde{\mathbf{H}}\mathbf{x} - \mathbf{y}\right\|_2^2 + \lambda_1 s(\mathbf{x}) + \lambda_2 \phi(\widetilde{\mathbf{H}}) \tag{5}$$

where $s : \mathbb{R}^d \to \mathbb{R}_{\geq 0}$ is a regularizer that returns a smaller value for a more probable estimate $\widehat{\mathbf{x}}$ (to ease notations, the training sample indexing $j$ is removed from $\widehat{\mathbf{x}}_j$ and $\mathbf{y}_j$). Also, $\phi : \mathbb{R}^{d \times d} \to \mathbb{R}_{\geq 0}$ is a regularizer that returns a smaller value for a more probable estimate for $\mathbf{H}$. The regularization parameters $\lambda_1 > 0$ and $\lambda_2 > 0$ determine the regularization strength of $s$ and $\phi$, respectively.

In usual inverse problems, the regularizer $s$ is set according to the type of the recovered data, e.g., natural images (Rudin et al., 1992). However, in our current problem of training data recovery, we can be more specific in the definition of the regularizer $s$ and use the knowledge that the data of interest was a training sample in the learning of the given autoencoder $f$.

Let us choose $s$ to be the regularizer that is implicitly induced by the trained autoencoder $f$. Specifically, the trained autoencoder $f$ is based on learning from its training samples. Furthermore, we assume that $f$ is overparameterized and its training allows it to overfit, and possibly to (numerically) perfectly fit, its training samples. Accordingly, there is a potential ability to use the trained autoencoder $f$ to understand if a given data sample was in its training dataset (for example, as explicitly done in membership inference of training data (Shokri et al., 2017; Carlini et al., 2022; Hu et al., 2022), although not necessarily for autoencoders). We rely on this potential ability to *implicitly* define $s_f : \mathbb{R}^d \to \mathbb{R}_{\geq 0}$ as a function that returns a lower value for an input that is more likely to be from the training dataset of the autoencoder $f$.

The regularizer $\phi$ for the estimate of $\mathbf{H}$ is defined based on the general assumptions on the degradation that was applied on the given training sample. In this paper, we focus on degraded training samples with many missing pixels. Namely, $\mathbf{H}$ is a pixel erasure operator, but the number and pattern (mask) of missing pixels is unknown in the recovery process. The accurate description of $\phi$ for our setting will be given later in this paper.

At this point, we are motivated to address the training data recovery task as the inverse problem in (5) with the regularizer $s_f$ that is induced by the trained autoencoder $f$ and a suitable $\phi$ for pixel erasure. Clearly, the optimization in (5) cannot be directly addressed due to having two intricate regularizers in its optimization cost and, more crucially, due to the implicit definition of the regularizer $s_f$ based on $f$. Remarkably, in the next section we develop a practical approach to solve this intricate inverse problem.

## 3 A PRACTICAL ALGORITHM FOR TRAINING DATA RECOVERY

Now we turn to develop a practical approach to address (5) with the regularizers $s_f$ and $\phi$. For a start, we decouple the joint optimization of $\mathbf{x}$ and $\widetilde{\mathbf{H}}$ via alternating minimization that yields an iterative procedure whose $t^{\text{th}}$ iteration includes the optimizations

$$\widehat{\mathbf{x}}^{(t)} = \underset{\mathbf{x} \in \mathbb{R}^d}{\arg \min} \left\| \widehat{\mathbf{H}}^{(t-1)} \mathbf{x} - \mathbf{y} \right\|_2^2 + \lambda_1 s_f(\mathbf{x}) \tag{6}$$

$$\widehat{\mathbf{H}}^{(t)} = \underset{\widetilde{\mathbf{H}} \in \mathbb{R}^{d \times d}}{\arg \min} \left\| \widetilde{\mathbf{H}} \widehat{\mathbf{x}}^{(t)} - \mathbf{y} \right\|_2^2 + \lambda_2 \phi(\widetilde{\mathbf{H}}). \tag{7}$$

Here, $\widehat{\mathbf{x}}^{(t)}$ and $\widehat{\mathbf{H}}^{(t)}$ are the estimates in the $t^{\text{th}}$ iteration (for $t = 1, 2, \dots$). The procedure needs to be initialized with some $\widehat{\mathbf{H}}^{(0)} \in \mathbb{R}^{d \times d}$ and to have a stopping criterion (see Appendix D.1).

Next, we explain how to address the optimizations (6) and (7) in Sections 3.1 and 3.2, respectively.

### 3.1 ESTIMATION OF A TRAINING SAMPLE $\mathbf{x}$ FOR A GIVEN DEGRADATION OPERATOR ESTIMATE

Let us address the optimization in (6) where a training sample is estimated using the implicit regularizer $s_f$ of a given trained autoencoder $f$, and for a degradation operator estimate $\widehat{\mathbf{H}}^{(t-1)}$. In this section we will develop a practical algorithm for this task using the alternating direction method of multipliers (ADMM) technique (Boyd et al., 2011; Afonso et al., 2010) and by extending the principle of plug and play priors (Venkatakrishnan et al., 2013; Sreehari et al., 2016) to autoencoders.

In this subsection we develop an iterative procedure to address (6). To avoid mixing the iteration indexing $t$ of the alternating minimization (6)-(7) with the to-be-defined iteration indexing $k$ of the ADMM procedure of (6), we use the following notations

$$\boldsymbol{\Theta} \triangleq \widehat{\mathbf{H}}^{(t-1)}, \quad \widehat{\boldsymbol{\xi}} \triangleq \widehat{\mathbf{x}}^{(t)}. \tag{8}$$

#### 3.1.1 THE ADMM FORM OF THE RECOVERY OPTIMIZATION

First, we will apply variable splitting on (6), and also use the notations in (8), to get

$$\{\widehat{\boldsymbol{\xi}}, \widehat{\mathbf{v}}\} = \underset{\mathbf{x}, \mathbf{v} \in \mathbb{R}^d}{\arg \min} \|\boldsymbol{\Theta} \mathbf{x} - \mathbf{y}\|_2^2 + \lambda_1 s_f(\mathbf{v}) \tag{9}$$

$$\text{subject to } \mathbf{x} = \mathbf{v}$$

where $\mathbf{v} \in \mathbb{R}^d$ is an auxiliary vector due to the split. Next, we develop (9) using an augmented Lagrangian and its corresponding iterative solution (of its scaled version) via the method of multipliers (Boyd et al., 2011, Ch. 2), whose $k^{\text{th}}$ iteration is

$$\{\widehat{\boldsymbol{\xi}}^{(k)}, \widehat{\mathbf{v}}^{(k)}\} = \underset{\mathbf{x}, \mathbf{v} \in \mathbb{R}^d}{\arg \min} \|\boldsymbol{\Theta} \mathbf{x} - \mathbf{y}\|_2^2 + \lambda_1 s_f(\mathbf{v}) + \frac{\gamma}{2} \left\| \mathbf{x} - \mathbf{v} + \mathbf{u}^{(k)} \right\|_2^2 \tag{10}$$

$$\mathbf{u}^{(k+1)} = \mathbf{u}^{(k)} + \left(\widehat{\boldsymbol{\xi}}^{(k)} - \widehat{\mathbf{v}}^{(k)}\right) \tag{11}$$

where $\mathbf{u}^{(k)} \in \mathbb{R}^d$ is the scaled dual-variable and $\gamma > 0$ is an auxiliary parameter that are introduced in the Lagrangian.

Then, applying one step of alternating minimization on (10) provides the iterative solution in the ADMM form (Boyd et al., 2011) whose $k^{\text{th}}$ iteration is

$$\widehat{\boldsymbol{\xi}}^{(k)} = \underset{\mathbf{x} \in \mathbb{R}^d}{\arg\min} \, \|\boldsymbol{\Theta}\mathbf{x} - \mathbf{y}\|_2^2 + \frac{\gamma}{2} \left\| \mathbf{x} - \widetilde{\mathbf{v}}^{(k)} \right\|_2^2 \tag{12}$$

$$\widehat{\mathbf{v}}^{(k)} = \underset{\mathbf{v} \in \mathbb{R}^d}{\arg\min} \, \lambda_1 s_f(\mathbf{v}) + \frac{\gamma}{2} \left\| \mathbf{v} - \widetilde{\boldsymbol{\xi}}^{(k)} \right\|_2^2 \tag{13}$$

$$\mathbf{u}^{(k+1)} = \mathbf{u}^{(k)} + \left(\widehat{\boldsymbol{\xi}}^{(k)} - \widehat{\mathbf{v}}^{(k)}\right) \tag{14}$$

where $\widetilde{\mathbf{v}}^{(k)} \triangleq \widehat{\mathbf{v}}^{(k-1)} - \mathbf{u}^{(k)}$ and $\widetilde{\boldsymbol{\xi}}^{(k)} \triangleq \widehat{\boldsymbol{\xi}}^{(k)} + \mathbf{u}^{(k)}$.

At this point, the main question is how to practically address the optimization in (13) that includes the implicitly-defined regularizer $s_f$.

### 3.1.2 Addressing the Optimization in (13) and its Implicit Regularizer $s_f$

To explain how to practically address (13), we will start by understanding this optimization task under a few assumptions that will be later relaxed. For a proper, closed, and convex function $s_f : \mathbb{R}^d \rightarrow \mathbb{R} \cup \{\infty\}$, the optimization in (13) is a Moreau proximity operator (Moreau, 1965; Hertrich et al., 2021; Sreehari et al., 2016). We will now turn to prove that there is a class of autoencoder architectures that correspond to Moreau proximity operators, and therefore correspond to the optimization in (13). Then, this association of (13) with some autoencoders will support our suggestion to replace (13) with a black-box application of *any* autoencoder (whose training data is asked to be recovered), regardless to whether it accurately corresponds to the optimization in (13).

A similar design philosophy is used in the plug-and-play priors approach (Venkatakrishnan et al., 2013; Sreehari et al., 2016) and its many applications (e.g., Dar et al., 2016; Rond et al., 2016; Brifman et al., 2016; Chan et al., 2017; Kamilov et al., 2017)) where a black-box image denoiser is applied within the ADMM iterations instead of solving an optimization that corresponds to a proximity operator for a regularization function that is implicitly defined by the denoiser.

For our current analysis of autoencoders, we start with a mathematical definition.

**Definition 2.** *A 2-layer tied autoencoder is defined for a weight matrix $\mathbf{W} \in \mathbb{R}^{m \times d}$ and componentwise activation function $\rho : \mathbb{R}^m \rightarrow \mathbb{R}^m$ as follows. The encoder is $f_{\text{enc}}(\mathbf{x}) = \rho(\mathbf{W}\mathbf{x})$ for $\mathbf{x} \in \mathbb{R}^d$, and the decoder is $f_{\text{dec}}(\mathbf{z}) = \mathbf{W}^T\mathbf{z}$ for $\mathbf{z} \in \mathbb{R}^m$. Accordingly, the entire autoencoder can be explicitly formulated as*

$$f(\mathbf{x}) = \mathbf{W}^T \rho(\mathbf{W}\mathbf{x}). \tag{15}$$

Definition 2 implies that a 2-layer tied autoencoder has a decoder weight matrix that is constrained to be the transpose of the encoder weight matrix. The next theorem is proved in Appendix A.

**Theorem 1.** *Any 2-layer tied autoencoder with*

- *a differentiable and componentwise activation function $\rho$ whose componentwise derivatives are in $[0, 1]$*

- *a weight matrix $\mathbf{W}$ whose singular values are in $[0, 1]$*

*is a Moreau proximity operator.*

**Corollary 1.1.** *If $f$ is a 2-layer tied autoencoder that satisfies the conditions of Theorem 1, there exists $s_f$ such that the optimization (13) corresponds to the application of $f$.*

Now, after we showed that some autoencoders correspond to optimization (13), we return to the practical setting where the given autoencoder $f$ can have various architectures. Similar to the plug-and-play priors approach (Venkatakrishnan et al., 2013; Sreehari et al., 2016) and its many applications (e.g., Dar et al., 2016; Rond et al., 2016; Brifman et al., 2016; Chan et al., 2017; Kamilov et al., 2017)), we suggest to replace the optimization in (13) by the application of the trained autoencoder on $\widetilde{\boldsymbol{\xi}}^{(k)}$, namely,

$$\widehat{\mathbf{v}}^{(k)} = f\left(\widetilde{\boldsymbol{\xi}}^{(k)}\right). \tag{16}$$

Although (16) does not necessarily accurately correspond to (13) for any autoencoder $f$, our experiments in Section 4 support this approach by showing the great empirical performance.

### 3.1.3 ADDRESSING THE PIXEL ERASURE DEGRADATION IN (12)

Let us examine the solution of optimization (12) in the ADMM procedure. This optimization does not involve the autoencoder $f$ nor its $s_f$, but it involves the (estimate of the) degradation operator. The degradation operator is represented by a $d \times d$ diagonal matrix $\boldsymbol{\Theta} \triangleq \widehat{\mathbf{H}}^{(t-1)}$ whose main diagonal values are either 0 or 1, indicating the positions of missing or available pixels, respectively. When $\boldsymbol{\Theta}$ is multiplied with $\mathbf{x}$, the resulting vector $\boldsymbol{\Theta}\mathbf{x}$ has its $i^{\text{th}}$ component either erased (set to 0) or unchanged based on the value of $\boldsymbol{\Theta}_{i,i}$. This structure of $\boldsymbol{\Theta}$ allows us to get the following simple, closed-form solution to the optimization in (12):

$$\widehat{\xi}_i^{(k)} = \begin{cases} \frac{y_i + \frac{\gamma}{2}\widetilde{v}_i^{(k)}}{1 + \frac{\gamma}{2}} & \text{if} \quad \boldsymbol{\Theta}_{i,i} = 1 \\ \widetilde{v}_i^{(k)} & \text{if} \quad \boldsymbol{\Theta}_{i,i} = 0 \end{cases} \tag{17}$$

where $\widehat{\xi}_i^{(k)}, y_i, \widetilde{v}_i^{(k)}$ are the $i^{\text{th}}$ components of the vectors $\widehat{\boldsymbol{\xi}}^{(k)}, \mathbf{y}, \widetilde{\mathbf{v}}^{(k)}$, respectively. More details on the development of (17) are provided in Appendix B.

The ADMM-based solution to (6), as developed in Section 3.1, is summarized in Algorithm 1.

---

**Algorithm 1** ADMM-based Solution to Optimization (6) for a Given Degradation Operator Estimate

---

1: Inputs: trained autoencoder $f$, degraded training sample $\mathbf{y}$, degradation operator estimate $\boldsymbol{\Theta}$, $\gamma$. Recall the notations of (8).
2: Initialize $k = 0, \hat{\mathbf{v}}^{(0)} = \mathbf{0}, \mathbf{u}^{(1)} = \mathbf{0}$.
3: **repeat**
4:     $k \leftarrow k + 1$
5:     $\widetilde{\mathbf{v}}^{(k)} \triangleq \hat{\mathbf{v}}^{(k-1)} - \mathbf{u}^{(k)}$
6:     $\widehat{\boldsymbol{\xi}}^{(k)} = \arg\min_{\mathbf{x} \in \mathbb{R}^d} \|\boldsymbol{\Theta}\mathbf{x} - \mathbf{y}\|_2^2 + \frac{\gamma}{2} \|\mathbf{x} - \widetilde{\mathbf{v}}^{(k)}\|_2^2$   (see the closed form solution in (17))
7:     $\widetilde{\boldsymbol{\xi}}^{(k)} \triangleq \widehat{\boldsymbol{\xi}}^{(k)} + \mathbf{u}^{(k)}$
8:     $\widehat{\mathbf{v}}^{(k)} = f(\widetilde{\boldsymbol{\xi}}^{(k)})$
9:     $\mathbf{u}^{(k+1)} = \mathbf{u}^{(k)} + \left( \hat{\boldsymbol{\xi}}^{(k)} - \hat{\mathbf{v}}^{(k)} \right)$
10: **until** stopping criterion is satisfied
11: Output: The training sample estimate $\widehat{\boldsymbol{\xi}}^{(k)}$ from the last iteration, this should be used as the solution $\widehat{\mathbf{x}}^{(t)}$ to the optimization in (6).

---

### 3.2 ESTIMATION OF A DEGRADATION OPERATOR FOR A GIVEN TRAINING SAMPLE ESTIMATE

Recall that our recovery problem has the alternating minimization form of (6)-(7). In Section 3.1 we showed how to address (6) via ADMM, which by itself is an iterative procedure (see Algorithm 1). Now we turn to address the second optimization in the alternating minimization, namely, the estimation of the degradation operator in (7).

In this paper, we consider degradations that erase many pixels from the original images used for training the autoencoder $f$. Therefore, the degradation operator $\mathbf{H}$ is a $d \times d$ diagonal matrix whose main diagonal includes zeros and ones only. We assume that this general structure is known and can be used in the recovery procedure, although without knowing the number and pattern of missing pixels.

According to the assumption on the pixel erasure degradation, we define the regularizer $\phi$ as the indicator function

$$\phi(\widetilde{\mathbf{H}}) = \begin{cases} 0 & \text{if } \widetilde{\mathbf{H}} \text{ is a diagonal matrix with } \widetilde{\mathbf{H}}_{i,i} \in \{0, 1\} \text{ for } i = 1, \dots, d \\ \infty & \text{otherwise.} \end{cases} \tag{18}$$

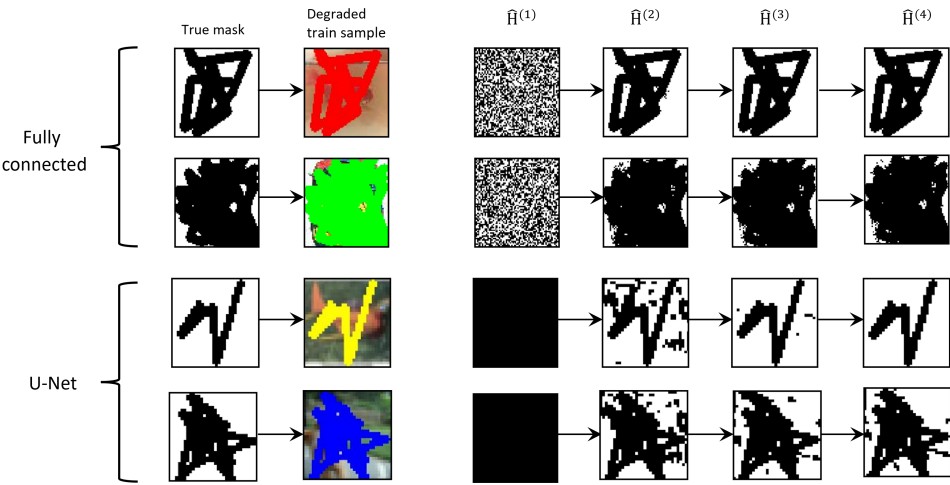

Figure 2: The estimation of $\widehat{\mathbf{H}}$ along the iterations. $\widehat{\mathbf{H}}^{(t)}$ is the estimate of $\mathbf{H}$ at the $t^{\text{th}}$ iteration of the proposed recovery algorithm via alternating minimization (6)-(7). Here, the diagonal matrix $\widehat{\mathbf{H}}^{(t)}$ is shown as an image.

This indicator function of $\phi$ forces the estimate $\widehat{\mathbf{H}}^{(t)}$ in (7) to be a diagonal matrix with zeros and ones on its main diagonal. Note that this indicator function implies that the specific positive value of $\lambda_2$ does not matter.

Using the definition of $\phi$ in (18), we can write the optimization (7) as $\widehat{\mathbf{H}}^{(t)} = \text{diag}\{\widehat{\mathbf{H}}_{1,1}^{(t)}, \ldots, \widehat{\mathbf{H}}_{d,d}^{(t)}\}$ where

$$\{\widehat{\mathbf{H}}_{i,i}^{(t)}\}_{i=1}^d = \underset{\{\widetilde{\mathbf{H}}_{i,i}\}_{i=1}^d \in \{0,1\}}{\arg\min} \sum_{i:\ \widetilde{\mathbf{H}}_{i,i}=1} (\widehat{x}_i^{(t)} - y_i)^2 + \sum_{i:\ \widetilde{\mathbf{H}}_{i,i}=0} y_i^2. \tag{19}$$

Here $\widehat{x}_i^{(t)}$ and $y_i$ are the $i^{\text{th}}$ components ($i \in \{1, \ldots, d\}$) of the vectors $\widehat{\mathbf{x}}^{(t)}$ and $\mathbf{y}$, respectively, and $\widetilde{\mathbf{H}}_{i,i}$ is the $i^{\text{th}}$ component on the main diagonal of $\widetilde{\mathbf{H}}$. Then, under the assumption that $y_i \in [0, 1]$ for any $i$, the minimization in (19) can be solved by the diagonal matrix $\widehat{\mathbf{H}}^{(t)}$ whose $i^{\text{th}}$ component on the main diagonal is

$$\widehat{\mathbf{H}}_{i,i}^{(t)} = \begin{cases} 0 & \text{if } \widehat{x}_i^{(t)} > 2y_i \text{ or } \widehat{x}_i^{(t)} < 0 \\ 1 & \text{otherwise.} \end{cases} \tag{20}$$

The recovery procedure requires the initialization of $\widetilde{\mathbf{H}}^{(0)}$. For U-Net autoencoders, we set $\widetilde{\mathbf{H}}^{(0)} = \mathbf{0}$; otherwise, we set $\widetilde{\mathbf{H}}^{(0)} = \mathbf{H}_{\text{rand}}$ where $\mathbf{H}_{rand}$ is a diagonal matrix with main diagonal components that are 0 or 1, each of these values independently drawn with 50% chance. We determined the mask initialization form by training models on a validation set and evaluating the validation recovery performance.

At this point, we conclude the development of the iterative recovery algorithm via alternating minimization in (6)-(7): The optimization (6) is solved by (20), and optimization (7) is addressed via ADMM as in Algorithm 1. The stopping criteria for the iterations in our method are described in Appendix D.1.

## 4 EXPERIMENTAL RESULTS

In our first set of experiments, we examined two main autoencoder architectures: (i) a 10-layer fully connected (FC) network, which was examined for LReLU and PReLU activations; and (ii) a U-Net model (Ronneberger et al. (2015)) that was examined for LReLU, PReLU, and SoftPlus activations. The FC model was trained on a random balanced subset of 600 images across all classes from Tiny

ImageNet. The U-Net was trained on 50 random samples from the SVHN dataset. We trained the models up to perfectly fitting the data at an MSE train loss of $10^{-8}$. We also saved model checkpoints during training at higher train losses to evaluate the recovery ability at lower overfitting levels. For more details on the model architecture and recovery method implementation, see Appendices C and D, respectively.

We examine the recovery performance for the trained models based on the following three metrics. (i) *Accurate-recovery rate*: the percentage of training samples for which the MSE between the original and reconstructed images is less than $10^{-7}$ (i.e., the PSNR[1] is higher than 70dB). (ii) *Approximate-recovery rate*: the percentage of training samples for which the reconstruction MSE is less than $5 \cdot 10^{-4}$ (i.e., PSNR>33.01dB). (iii) *Average PSNR* of all the reconstructions of the training data.

We evaluated our proposed method, without the knowledge of the true missing pixel mask $\mathbf{H}$, and compared it to three alternatives: (a) Our proposed method but with knowing the true missing pixel mask $\mathbf{H}$, see details in Appendix E. (b) The simple iterative application of the trained autoencoder (AE) only, as was analyzed and shown by Radhakrishnan et al. (2020) to have good recovery performance in very specific settings. (c) A generic image inpainting method that utilizes the true mask of missing pixels. For this we use the Denoising Diffusion Null-space Method (DDNM) of Wang et al. (2023), which is a relatively recent technique with good inpainting performance also for ImageNet images. Note that alternatives (a) and (c) have the advantage of knowing the true pixel mask, which is unavailable for our proposed method and alternative (b).

Our method outperformed the three alternatives in recovery rate for both the accurate and approximate definitions. For accurate recovery, Figure 3 shows that our method outperformed the alternatives, e.g., for U-Net autoencoder at train loss of $10^{-8}$ and missing pixel mask 6, our proposed method (with unknown $\mathbf{H}$) achieves $78\%$ recovery, versus $4\%$ for the simple iterative autoencoding and $0\%$ for DDNM generic inpainting. Interestingly, sometimes our method succeeds to recover from models at higher train loss values (lower overfitting) where competing methods (b)-(c) have $0\%$ recovery rate (e.g., see results for masks 1,2,3 at train loss $10^{-7}$ in Figure 3). Yet, the results clearly show that a sufficiently high train loss (i.e., a sufficiently low or no overfitting) can prevent the recovery from all the examined methods. More evaluations are provided in Appendix F (specifically, the approximate-recovery rate and average PSNR evaluations appear in Figures F.1 and F.2). Unless otherwise specified, we consider pixel erasure without additive noise (i.e., $\sigma_\epsilon = 0$ in the degradation model (3)). Results for degradation with noise are provided in Appendix G.

To validate that our method's effectiveness stems from the autoencoder's induced regularization, we created Tiny ImageNet and SVHN test sets with 600 and 50 images respectively, matching the training set sizes. The most overfitted models (MSE of $10^{-8}$) were examined. We applied our method on to recovery test data from pixel erasure according to the same masks as in Figure F.1. For the test data, our method had $0\%$ accurate recovery rate across all architectures and masks, and $0\%$ approximate recovery rate for all except Mask 3 with the FC model which had 2/600 recovered images. These results demonstrate that our method is indeed for recovery of the specific training samples of a given autoencoder, and not just to generically recover the missing pixels from images that were not used in the training.

To evaluate our method in the regime of moderate overfitting, we trained autoencoders on larger datasets and up to MSE training loss of $10^{-4}$ (which is higher than the perfect fitting loss of $10^{-8}$ in our previous experiment). The two examined architectures in this experiment are: (i) A 20-layer FC autoencoder with Leaky ReLU activations, trained on a subset of 25,000 images from Tiny ImageNet. (ii) A U-Net autoencoder with Softplus activations, trained on 1000 CIFAR-10 images. Tables 1-2 shows that our method achieved higher approximate-recovery performance compared to the simple iterative autoencoder application. This further establishes the significant improved recovery that our method achieves.

## 5 CONCLUSION

In this work, we have established a new perspective on the recovery of training data from overparameterized autoencoders. We defined an inverse problem with a regularizer that is implicitly defined based on the trained autoencoder. We addressed this problem using the ADMM optimization technique

---

[1]In our case where the original image pixel values are in $[0, 1]$: $PSNR = 10 \log_{10}\left(\frac{1}{MSE}\right)$

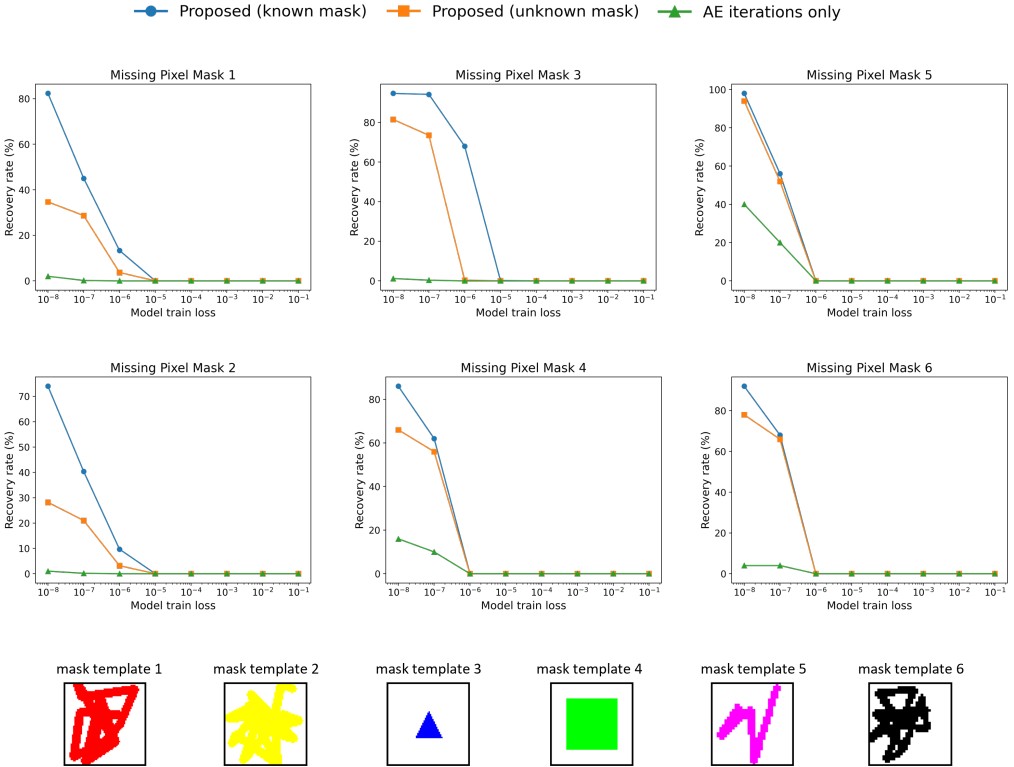

Figure 3: **Accurate recovery rates** for recovery from degradation due to various missing pixel masks. The results are obtained from three different architectures (each at various train loss values): Masks 1 and 2 were examined for a 10-layer FC autoencoder with LReLU activations; Mask 3 was examined for a 10-layer FC autoencoder with PReLU activations; Masks 4, 5, 6 were examined for the U-Net architecture with LReLU, softplus, and PReLU activations, respectively. The evaluated recovery methods are the proposed method for unknown mask (orange curves), the proposed method for a known mask (blue curves), the simple iterations of the autoencoder only (green curves), and the generic inpainting method DDNM that got 0% accurate-recovery rate for all these settings and therefore is not graphically shown.

| Missing pixel mask | ■ | ▧ | ▧ |
|---|---|---|---|
| AE iterations only | 7.64% | 0.78% | 1.92% |
| Proposed method (unknown H) | 54.2% | 45.2% | 56.9% |
| Proposed method (known H) | 80.6% | 75.9% | 84.2% |

Table 1: Approx.-recovery rates for 20-layer FC (25k Tiny-ImageNet training samples)

| Missing pixel mask | ▧ | ▧ | ▧ |
|---|---|---|---|
| AE iterations only | 0.6% | 10.2% | 1.2% |
| Proposed method (unknown H) | 50.3% | 75.9% | 22.9% |
| Proposed method (known H) | 59.3% | 86.2% | 28.8% |

Table 2: Approx.-recovery rates for U-Net (1000 CIFAR-10 training samples)

and an extension of the plug and play priors idea, and developed a practical iterative method for the recovery of training data based on applications of the trained autoencoder.

Our new approach significantly improves the training data recovery performance, compared to previous methods. We showed for a variety of settings that our method is able to recover up to tens of percents more of the training dataset, and to operate well also on models with moderate overfitting (unlike previous methods that require a stricter overfitting, or perfect fitting). Specifically, our results show that various settings do not maintain the privacy of their training data as can be understood from the results of previous methods. More generally, we believe that our work provides useful study cases and insights that can pave the way for a better understanding of memorization phenomena and how to design autoencoders that keep the privacy of their training data.

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

## A   PROOF OF THEOREM 1

In this Appendix, we prove Theorem 1.

**Lemma A.1.** *Given a 2-layer tied autoencoder, $f$, which can be formulated as*

$$f(\mathbf{x}) = \mathbf{W}^T \rho(\mathbf{W}\mathbf{x})$$

*for $\mathbf{x} \in \mathbb{R}^d$, where $\mathbf{W} \in \mathbb{R}^{m \times d}$, $\rho : \mathbb{R}^m \to \mathbb{R}^m$. Specifically, the activation function has a (separable) componentwise form*

$$\rho(\mathbf{z}) = [\bar{\rho}(z_1), \ldots, \bar{\rho}(z_m)]^T \quad \text{for } \mathbf{z} \in \mathbb{R}^m \tag{21}$$

*whose $j^{\text{th}}$ component is denoted as $z_j$, and for a scalar activation function $\bar{\rho} : \mathbb{R} \to \mathbb{R}$. We denote $\mathbf{z} \triangleq \mathbf{W}\mathbf{x}$.*

*Then, the Jacobian matrix of $f$ is*

$$\frac{df(\mathbf{x})}{d\mathbf{x}} = \mathbf{W}^T \text{diag}\left( \frac{d\bar{\rho}(z_1)}{dz_1}, \frac{d\bar{\rho}(z_2)}{dz_2}, \ldots, \frac{d\bar{\rho}(z_m)}{dz_m} \right) \mathbf{W}. \tag{22}$$

*where $diag(\cdot)$ represents a diagonal matrix with the given components along the main diagonal.*

*Proof.* Let us define auxiliary variables. In addition to $\mathbf{z} \triangleq \mathbf{W}\mathbf{x}$, we also define $\mathbf{a} \triangleq \rho(\mathbf{z})$ and $\boldsymbol{\xi} \triangleq f(\mathbf{x}) = \mathbf{W}^T \mathbf{a}$.

Then, by the chain rule, we get

$$\frac{df(\mathbf{x})}{d\mathbf{x}} = \frac{d\boldsymbol{\xi}}{d\mathbf{a}} \cdot \frac{d\mathbf{a}}{d\mathbf{z}} \cdot \frac{d\mathbf{z}}{d\mathbf{x}} = \mathbf{W}^T \cdot \frac{d\mathbf{a}}{d\mathbf{z}} \cdot \mathbf{W} \tag{23}$$

Next, by definition, $\frac{d\mathbf{a}}{d\mathbf{z}} = \frac{d\rho(\mathbf{z})}{d\mathbf{z}}$, and since $\rho(\mathbf{z})$ is a vector of componentwise activation functions (see (21)), this Jacobian is a $m \times m$ diagonal matrix in the form of

$$\frac{d\rho(\mathbf{z})}{d\mathbf{z}} = \text{diag}\left( \frac{d\bar{\rho}(z_1)}{dz_1}, \frac{d\bar{\rho}(z_2)}{dz_2}, \ldots, \frac{d\bar{\rho}(z_m)}{dz_m} \right) \tag{24}$$

Substituting (24) back into (23) gives the Jacobian formula of (22). $\qquad\square$

**Corollary A.1.** *Let $\bar{\rho} : \mathbb{R} \to \mathbb{R}$ be a scalar activation function that is differentiable and has derivatives in $[0, 1]$, namely, $\frac{d\bar{\rho}(z)}{dz} \in [0, 1]$ for any $z \in \mathbb{R}$. Then, for such activation function, a 2-layer tied autoencoder has a Jacobian in the form of $\mathbf{W}^T \mathbf{D} \mathbf{W}$, where $\mathbf{D}$ is a diagonal matrix whose values are in $[0, 1]$.*

**Lemma A.2.** *Let* $\mathbf{W} \in \mathbb{R}^{c_2 \times c_1}$ *and* $\mathbf{D}$ *is a* $c_2 \times c_2$ *diagonal matrix with values in* $[0, 1]$*. Then,* $\mathbf{W}^T \mathbf{D} \mathbf{W}$ *is a symmetric positive semi-definite matrix.*

*Proof.* First, we show that the matrix is symmetric:

$$(\mathbf{W}^T \mathbf{D} \mathbf{W})^T = \mathbf{W}^T \mathbf{D}^T \mathbf{W} = \mathbf{W}^T \mathbf{D} \mathbf{W},$$

where $\mathbf{D}^T = \mathbf{D}$ due to the symmetry of a diagonal matrix.

Now, we prove that the matrix $\mathbf{W}^T \mathbf{D} \mathbf{W}$ is positive semi-definite. Namely, we need to show that for any $\mathbf{r} \in \mathbb{R}^{c_1}$, $\mathbf{r}^T \mathbf{W}^T \mathbf{D} \mathbf{W} \mathbf{r} \geq 0$. Define $\widetilde{\mathbf{r}} \triangleq \mathbf{W}\mathbf{r}$, then we need to show that $\widetilde{\mathbf{r}}^T \mathbf{D} \widetilde{\mathbf{r}} \geq 0$. This holds because, by denoting $\widetilde{r}_i$ as the $i^{\text{th}}$ component of $\widetilde{\mathbf{r}}$ and $\mathbf{D}_{i,i}$ as the $i^{\text{th}}$ main diagonal component of $\mathbf{D}$, we get $\widetilde{\mathbf{r}}^T \mathbf{D} \widetilde{\mathbf{r}} = \sum_{i=1}^{c_1} \mathbf{D}_{i,i} \widetilde{r}_i^2 \geq 0$ because $\mathbf{D}_{i,i} \in [0, 1]$ for any $i$. $\qquad \square$

Now we proceed to prove Theorem 1, i.e., that a tied autoencoder $f$ from the class described in the theorem is a Moreau proximity operator.

*Proof.* We prove that $f(\mathbf{x})$ is a Moreau proximity operator by showing that the Jacobian matrix of $f(\mathbf{x})$ w.r.t. any $\mathbf{x} \in \mathbb{R}^d$ satisfies two properties: (i) the Jacobian is a symmetric matrix, and (ii) all the Jacobian matrix eigenvalues are real and in the range of $[0, 1]$. Note that previous works on plug and play prior used conditions (i)-(ii) to prove that special types of denoisers are Moreau proximity operators, for example, see (Sreehari et al., 2016).

Consider a 2-layer tied autoencoder, $f$, which can be formulated as

$$f(\mathbf{x}) = \mathbf{W}^T \rho(\mathbf{W}\mathbf{x})$$

where $\mathbf{W} \in \mathbb{R}^{m \times d}$ has all its singular values in $[0, 1]$, and $\rho : \mathbb{R}^m \to \mathbb{R}^m$ is a componentwise activation function as in (21) that is based on a differentiable scalar activation function $\bar{\rho} : \mathbb{R} \to \mathbb{R}$ whose derivative is in $[0, 1]$.

We will now prove that $f$ is a Moreau proximity operator.

From Corollary A.1 and Lemma A.2, we get that $f$ has a Jacobian matrix $\mathbf{W}^T \mathbf{D} \mathbf{W}$, which is symmetric and positive semi-definite.

We will now prove that the eigenvalues of $\mathbf{W}^T \mathbf{D} \mathbf{W}$ are all in $[0, 1]$. First, notice that the singular values of $\mathbf{D}$ are the same as the eigenvalues, which are the diagonal elements that are in $[0, 1]$. In addition, the singular values of $\mathbf{W}^T$ are the same as the singular values of $\mathbf{W}$, which are in $[0, 1]$ by the assumption of Theorem 1. Hence, the singular values of each of the matrices in the product $\mathbf{W}^T \mathbf{D} \mathbf{W}$ are in $[0, 1]$. It is also well known that for every two matrices, $\mathbf{A} \in \mathbb{R}^{q_1 \times q_2}$, $\mathbf{B} \in \mathbb{R}^{q_2 \times q_3}$,

$$\sigma_i(\mathbf{A}\mathbf{B}) \leq \sigma_1(\mathbf{A})\sigma_i(\mathbf{B}) \tag{25}$$

where $\sigma_i$ denotes the $i^{\text{th}}$ largest singular value of a corresponding matrix. Hence, for $\mathbf{C} \in \mathbb{R}^{q_3 \times q_4}$,

$$\sigma_i(\mathbf{A}\mathbf{B}\mathbf{C}) \leq \sigma_1(\mathbf{A})\sigma_1(\mathbf{B})\sigma_i(\mathbf{C}).$$

In our case, $\sigma_1(\mathbf{W}^T) \leq 1$, $\sigma_1(\mathbf{D}) \leq 1$, $\sigma_i(\mathbf{W}) \leq 1$, and therefore

$$\sigma_i(\mathbf{W}^T \mathbf{D} \mathbf{W}) \leq \sigma_1(\mathbf{W}^T)\sigma_1(\mathbf{D})\sigma_i(\mathbf{W}) \leq 1. \tag{26}$$

Consequently, all the singular values of the Jacobian matrix are in $[0, 1]$. Moreover, for real symmetric matrices, the absolute values of the eigenvalues are equal to the singular values. Since the Jacobian of our 2-layer tied autoencoder is real and symmetric, by (26) we get that the eigenvalues of this Jacobian are in $[-1, 1]$. Moreover, by Lemma A.2, the Jacobian is symmetric positive semi-definite and therefore its eigenvalues are non-negative; accordingly, all the eigenvalues of the Jacobian are in $[0, 1]$.

To sum up, we showed that a 2-layer tied autoencoder that satisfies the conditions in Theorem 1 has a symmetric semi-positive definite Jacobian with eigenvalues in $[0, 1]$; therefore, such a 2-layer autoencoder is a Moreau proximity operator.

$\qquad \square$

## B    PROOF OF EQUATION (17)

Recall the notations in (8). The optimization problem (17), i.e.,

$$\widehat{\boldsymbol{\xi}}^{(k)} = \arg\min_{\mathbf{x} \in \mathbb{R}^d} \|\boldsymbol{\Theta}\mathbf{x} - \mathbf{y}\|_2^2 + \frac{\gamma}{2} \left\|\mathbf{x} - \widetilde{\mathbf{v}}^{(k)}\right\|_2^2 \tag{27}$$

has a closed form solution

$$\widehat{\boldsymbol{\xi}}^{(k)} = \left(\boldsymbol{\Theta}^T\boldsymbol{\Theta} + \frac{\gamma}{2}\mathbf{I}\right)^{-1} \left(\boldsymbol{\Theta}\mathbf{y} + \frac{\gamma}{2}\widetilde{\mathbf{v}}^{(k)}\right) \tag{28}$$

Then, the diagonal structure of $\boldsymbol{\Theta}$ with zeros and ones on its main diagonal implies that $\boldsymbol{\Theta}^T\boldsymbol{\Theta} = \boldsymbol{\Theta}$ and, therefore,

$$\widehat{\boldsymbol{\xi}}^{(k)} = \left(\boldsymbol{\Theta} + \frac{\gamma}{2}\mathbf{I}\right)^{-1} \left(\boldsymbol{\Theta}\mathbf{y} + \frac{\gamma}{2}\widetilde{\mathbf{v}}^{(k)}\right) \tag{29}$$

that can be further simplified to the componentwise form of

$$\widehat{\xi}_i^{(k)} = \begin{cases} \widetilde{v}_i^{(k)}, & \text{if } \boldsymbol{\Theta}_{i,i} = 0 \\ \frac{y_i + \frac{\gamma}{2}\widetilde{v}_i^{(k)}}{1 + \frac{\gamma}{2}}, & \text{if } \boldsymbol{\Theta}_{i,i} = 1 \end{cases}$$

where $\widehat{\xi}_i^{(k)}$, $y_i$, $\widetilde{v}_i^{(k)}$ are the $i^{\text{th}}$ components of the vectors $\widehat{\boldsymbol{\xi}}^{(k)}$, $\mathbf{y}$, $\widetilde{\mathbf{v}}^{(k)}$, respectively.

## C    THE EXAMINED AUTOENCODER ARCHITECTURES

We trained two fully connected (FC) autoencoder architectures, one with 10 layers and one with 20 layers. We also trained a U-Net autoencoder model. The activation functions used for training the models were Leaky ReLU, PReLU, and Softplus. To ensure reproducibility, all experiments were with seed 42.

### C.1    PERFECT FITTING REGIME

In the experiments that arrive to the perfect fitting regime, the FC models were trained on 600 images from Tiny ImageNet (at $64 \times 64 \times 3$ pixel size) up to a minimum MSE loss of $10^{-8}$, which can be considered as numerical perfect fitting. During training, intermediate models at higher train loss values were saved and used later for evaluation of the recovery at lower overfitting levels (see, e.g., Figure F.1). We trained two versions of the 10-layer and 20-layer fully connected models using Leaky ReLU and PReLU activations for each architecture.

The U-Net model was trained on 50 images from the SVHN dataset (at $32 \times 32 \times 3$ pixel size) also to an MSE train loss of $10^{-8}$, while saving intermediate models at higher train loss values. We examined U-Net architectures for three different activation functions: Leaky ReLU, PReLU, and Softplus.

### C.2    MODERATE OVERFITTING REGIME

In the moderate overfitting regime, we trained a 20-layer FC model on a larger subset of 25,000 images from Tiny ImageNet (at $64 \times 64 \times 3$ pixel size) with Leaky ReLU activations to a loss of $10^{-4}$. This achieves moderate overfitting, yet not perfect fitting of the training data.

The U-Net model was trained on 1000 images from the CIFAR-10 dataset (at $32 \times 32 \times 3$ pixel size) to an MSE train loss of $10^{-4}$. We examined U-Net architectures for three different activation functions: Leaky ReLU, PReLU, and Softplus.

## D    THE PROPOSED METHOD: ADDITIONAL IMPLEMENTATION DETAILS

### D.1    STOPPING CRITERION OF THE PROPOSED METHOD

The stopping criterion for the ADMM via Algorithm 1 (which solves equation (6)) is a predefined number of iterations. We set this to 40 iterations. Each alternating minimization iteration in our overall algorithm includes one ADMM procedure.

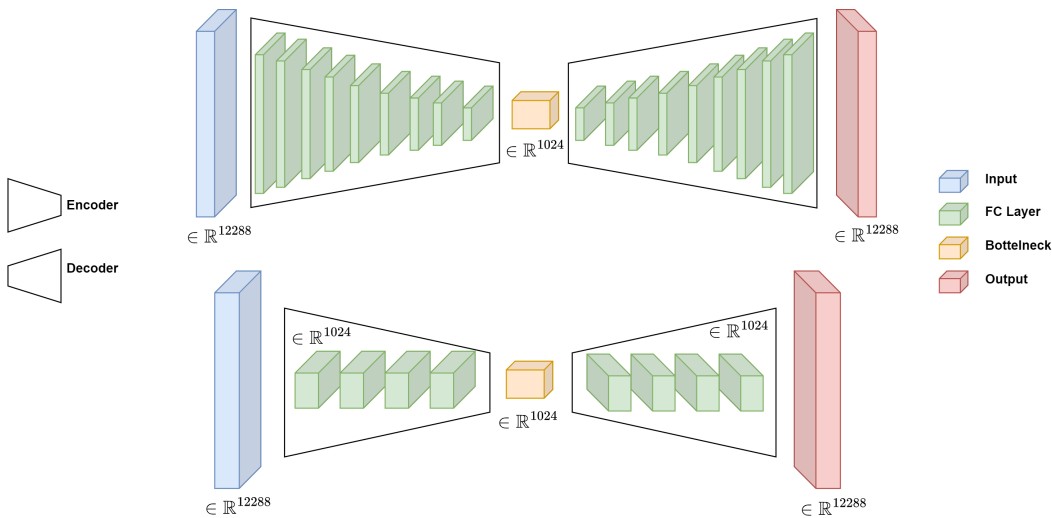

Figure C.1: Architecture of 10 layers and 20 layers fully connected autoencoders for the Tiny ImageNet dataset (a subset of images, at $64 \times 64 \times 3$ pixel size).

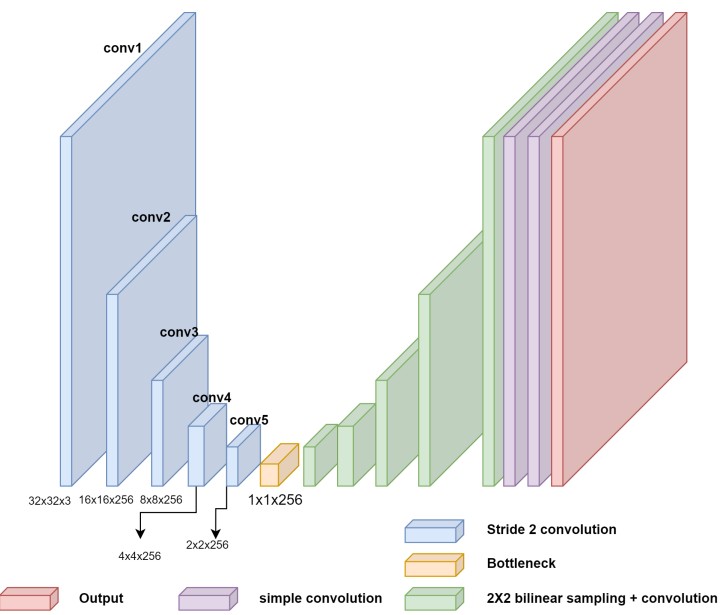

Figure C.2: Architecture of U-Net autoencoder for the CIFAR-10 and SVHN datasets (subsets of images, at $32 \times 32 \times 3$ pixel size). stride and padding are 1, kernel size is $3 \times 3$, and an activation function is applied after every convolution.

The stopping criterion for the entire recovery algorithm via alternating minimization, (6)-(7), is that the MSE between successive estimates $\widehat{\mathbf{x}}^{(t)}$ must be below a threshold for 3 consecutive iterations; we set this MSE threshold to $10^{-9}$.

## D.2 $\gamma$ VALUES FOR THE PROPOSED METHOD

The $\gamma$ value of the ADMM (Algorithm 1) was set as follows. $\gamma = 0.5$ for the 10-layer FC autoencoder with LReLU activations. $\gamma = 0.1$ for the 10-layer FC autoencoder with PReLU activations, and for the 20-layer FC autoencoder. $\gamma = 1$ for the U-Net architecture.

## E    THE SPECIAL CASE OF THE PROPOSED METHOD FOR A KNOWN $\mathbf{H}$

In this appendix, we will briefly discuss the case where the degradation operator $\mathbf{H}$ is known. In this paper we focus on the case where $\mathbf{H}$ is unknown, and use the special case of a known $\mathbf{H}$ for the purpose of performance comparison.

Recall the inverse problem (5) for the general case where $\mathbf{H}$ is unknown. If $\mathbf{H}$ is known, (5) can be reduced into the form of

$$\widehat{\mathbf{x}} = \underset{\mathbf{x} \in \mathbb{R}^d}{\arg\min} \| \mathbf{H}\mathbf{x} - \mathbf{y} \|_2^2 + \lambda_1 s_f(\mathbf{x}). \tag{30}$$

This problem has the same form as the optimization (6) from the alternating minimization procedure, but here in (30) we have the true $\mathbf{H}$ instead of its estimate. Accordingly, we can address the optimization in (30) via Algorithm 1 with $\mathbf{\Theta} = \mathbf{H}$. Since we know $\mathbf{H}$, and if we also know that the degradation is pixel erasure without additive noise (i.e., $\sigma_\epsilon = 0$ in the degradation model (3)) we improve the output of Algorithm 1 by setting the known (non-erased) pixels from $\mathbf{y}$ in the estimate $\widehat{\mathbf{x}}$, i.e.,

$$\widehat{\mathbf{x}}^{\text{known } \mathbf{H}} = (\mathbf{I}_d - \mathbf{H})\widehat{\mathbf{x}} + \mathbf{H}\mathbf{y}. \tag{31}$$

## F    ADDITIONAL EXPERIMENTAL RESULTS: ANALYSIS OF THE EXPERIMENTS OF FIGURE 3 USING OTHER PERFORMANCE METRICS

Figure 3 shows the recovery performance in terms of *accurate recovery* rates, where the threshold for considering a successful recovery is MSE of $10^{-7}$ (PSNR 70dB) between the estimated and original images. Here, in Figure F.1 shows the corresponding *approximate recovery* rates, where the threshold for considering a successful recovery is MSE of $5 \cdot 10^{-4}$ (PSNR 33.01dB) between the estimated and original images. Moreover, in Figure F.2 shows the corresponding *average PSNR* of all the estimates (i.e., the averaging is over all the samples in the training dataset and not only those who are considered as recovered by one of the specified thresholds).

To further understand the effect of the recovery threshold definition, in Figure F.3 we show the recovery rate behavior of the training images as the for several recovery threshold definitions between MSE values of $10^{-3}$ and $10^{-6}$.

The definition of the approximate recovery rate based on MSE of $5 \cdot 10^{-4}$ (PSNR 33.01dB) in Figure F.1 shows that our proposed method significantly outperforms the previous approach of autoencoder iterations only (see, for example, results for masks 1,2,4 in Figure F.1). However, this definition of approximate recovery also introduces an artifact in the form of some approximate recovery percentages for the generic inpainting (DDNM) method (see, for example, the results for masks 3,5,6 in Figure F.1), which can be explained by the corresponding average PSNR values in Figure F.2 (see, for example, the results for masks 3,5,6 in Figure F.2 where the red lines show that the generic inpainting DDNM has average PSNR near the approximate recovery threshold PSNR of 33.01). For this reason, we focus on the accurate recovery rate performance that is shown in Figure 3 and also suggest to examine the impressive performance gain of our method in terms of average PSNR in Figure F.2.

## G    ADDITIONAL EXPERIMENTAL RESULTS: DEGRADATION OF PIXEL ERASURE WITH ADDITIVE NOISE

In this appendix section, we evaluate an experiment with additive noise in addition to the degradation operator. Recall the degradation model (3) and note that here we have noise standard deviation of $\sigma_\epsilon = 0.02$. The degradation operator, $\mathbf{H}$, is based on mask 1 that appears, e.g., in Figure 3. Our method uses here $\gamma = 0.1$.

Figure G.1 demonstrates that, also when additive Gaussian noise is added to the pixel erasure degradation, the proposed method significantly improves the recovery performance compared to the simple iterations-only approach.

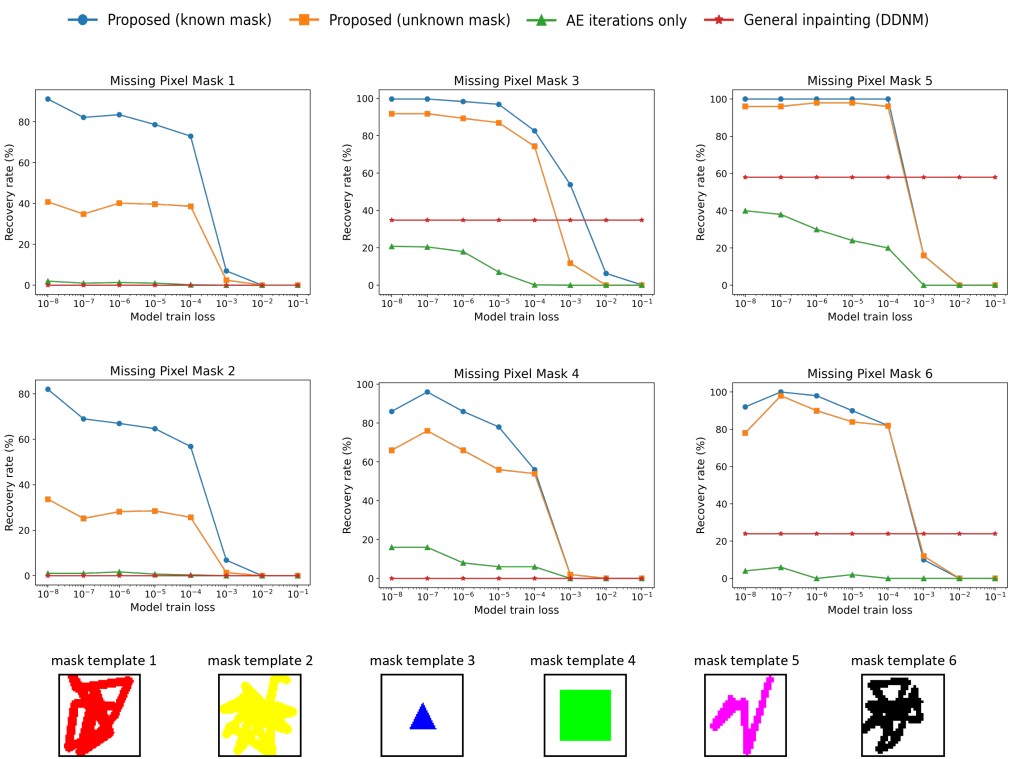

Figure F.1: **Approximate recovery rate** results. Besides the definition of the recovery rate threshold (and the visual presentation of the generic inpainting DDNM performance in red curves), all the other setting details are as in Figure 3.

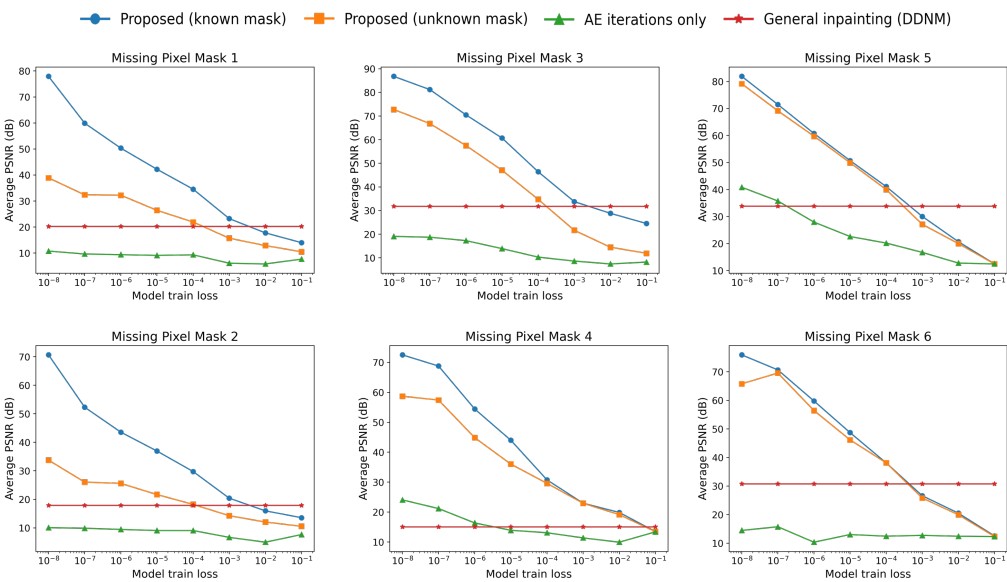

Figure F.2: **Average PSNR** results for degraded image reconstruction corresponding to Figure 3 and Figure F.1. The average is computed over the recovery attempts of all the training dataset. All the other setting details are as in Figure 3.

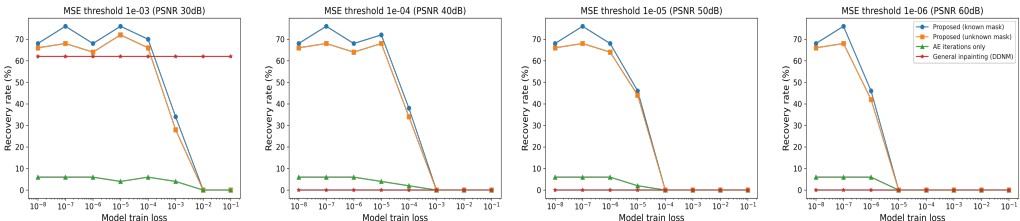

Figure F.3: The recovery rate results according to several MSE thresholds. These results correspond to the experiment on the U-Net architecture and mask 6.

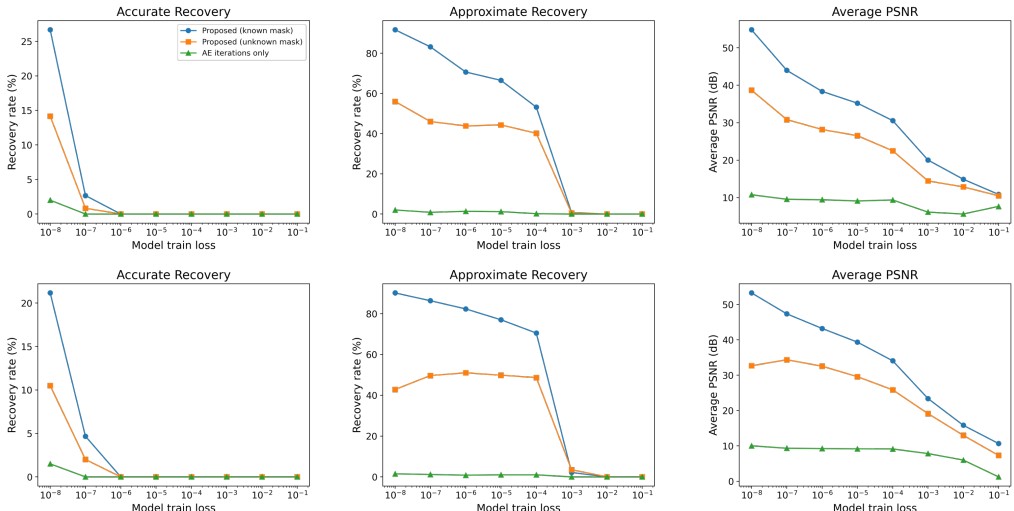

Figure G.1: Recovery results of degraded samples with additive noise ($\sigma_\epsilon = 0.02$). The results are of two architectures, 10-layer (first row) and 20-layer (second row) fully-connected autoencoder with LReLU activations.

