# OpenReview forum: "Recovery of Training Data from Overparameterized Autoencoders: An Inverse Problem Perspective"
_ICLR.cc/2024/Conference — ICLR 2024 Conference Withdrawn Submission_

### Official Review · Reviewer_VRp5 · 2023-10-23

**Soundness:** 3 good
**Presentation:** 3 good
**Contribution:** 2 fair
**Rating:** 6
**Confidence:** 3

**Summary:**

The paper proposes a novel approach for recovering images on which an auto-encoder was trained. The method assumes access to a set of degraded training images. Specifically, the degrading is done via a noisy *linear* operator. The authors study a particular variation of such an operator which erases the image pixels (i.e., diagonal matrix with $\\{0,1\\}$ entries). On a high-level, the method consists iof alternating steps of estimating the image $\hat{\mathrm{x}}$ and degrading operator $\hat{\mathrm{\boldsymbol{H}}}$ via an ADMM-like algorithm. The authors demonstrate the superiority of  their approach by comparing with DDNM method and iterative application of trained autoencoder (Radhakrishnan et al.), while also validating a variation of their method which has an access to the true pixel mask $\hat{\mathrm{\boldsymbol{H}}}$.

**Strengths:**

- a novel method for image recovery, that seems to be empirically superior to the existing alternatives
- the methodology is based on a well-established ADMM method

**Weaknesses:**

- the type of degrading of the images is limited to noisy linear ones
- the method still seem to assume a particular structure of $\hat{\mathrm{\boldsymbol{H}}}$ (i.e., diagonal for pixel erasure), e.g., the choice of the regularizer $\phi$ and etc.
- having access to a degraded training samples for recovery is a bit less practical than one can imagine
- the comparison and overall experimental evaluation seems a bit lacking

**Questions:**

- do the authors think that their method can potentially treat a general form of unknown $\hat{\mathrm{\boldsymbol{H}}}$? If so, I would be delighted to see some numerical evidence for that, even less rigorous would do

- it would be interesting to see, if the method still performs well under a mismatched scenario, i.e., the degrading process itself is not linear, but one can assume a certain form of $\hat{\mathrm{\boldsymbol{H}}}$ that replicates it close enough

- It would be interesting to see whether the method is able to perform well on an image which is not used during the training but close enough: pick some simple dataset and subsample images of a certain class and than look at the performance of the unused remaining ones

---

### Official Review · Reviewer_1EzY · 2023-10-29

**Soundness:** 2 fair
**Presentation:** 3 good
**Contribution:** 2 fair
**Rating:** 3
**Confidence:** 3

**Summary:**

The paper proposed an ADMM algorithm to recover training data from degraded observations. In detail, this paper looks at masking-based linear degradation functions similar to those in linear inverse problems, for AEs that can almost perfectly fit the training data. The proposed method outperforms previous techniques and improves recovery performance. The experiments also show a strong correlation between overfitting and recovery performance of the proposed method.

**Strengths:**

- The paper is clearly written and easy to follow. The formulation follows naturally from the reconstruction task, and all algorithm design follows naturally from the training objective. The decomposition of all parts of the algorithms is also straightforward.
- It is an interesting application of ADMM to data reconstruction tasks and can be potentially used for the general neural network architectures with the plug-and-play method. There is theory provided to support this change.
- The experimental results show improvement over a number of baselines, and even under the noisy degradation case.

**Weaknesses:**

- The motivation of this paper is unclear. The paper mentions privacy challenges in modern ML at the beginning of the paper, but there lacks connection between the proposed data reconstruction technique and realistic privacy challenges. There should be more discussion (and examples) on how this formulation can represent realistic privacy concerns. For instance, one speculative example can be generating sensitive information with appropriate prompts in an LLM, which seems to be related to masking degradation. Another example can be image copyright issues, which further establishes a connection to membership inference. I hope the authors can draw a connection between the proposed formulation and these practical challenges, and make a clear statement on the attack taxonamy (e.g. black- or white-box, number of queries, etc.).
- The word "overparameterized" is not appropriate in my thoughts. It usually means the middle hidden layer is much wider than input/output dimensions, which is in contrast to AEs where the latent dimension is usually smaller. Overparameterized networks may not always interpolate training data; they usually do under sgd or gd, but not under some other optimizers; and extrapolation may happen together with interpolation, which indicates non-overfitting. Based on your assumption, it is more accurate to use words like interpolating or overfitted AEs.
- While there is theoretical justification for using plug-and-play to avoid the explicit definition of $s_f$, there isn't convergence analysis on the proposed algorithm. The initialization selection also seems to be heuristic. It is therefore reasonable to doubt how robust the proposed algorithm is, as well as its potential to generalize to a wider range of problems.
- There isn't a proposed algorithm for noisy degradation that leverages $\sigma_{\epsilon}$ assuming it's known, which limits the scope the proposed task. While the experiments for small $\sigma_{\epsilon}$ shows the scalability of the proposed method, it is possible to fail for larger $\sigma_{\epsilon}$, which is not discussed in the paper.
- The experiments are only conducted on very small training sets. From my understanding this is to ensure that the model overfits. However, it is very far away from practical scenarios, where most ML models under trustworthy concerns are giant and trained on massive data. It would be more interesting to look at (pretrained) models trained on the full datasets. These models can overfit on parts of the training set and not overfit on others; this can help us better understand the effect of overfitting to reconstruction.
- There is no discussion on what will happen if you input a degraded sample $\notin$ training set but close to some $x_i$.

**Questions:**

The questions correspond to the weakness mentioned above.
- Motivation and connection to realistic privacy concerns?
- Any convergence and initialization analysis?
- Is there an algorithm for noisy degradation?
- What is the maximum noise for the proposed method to perform well?
- Any experiments for standard full training sets?
- What does the proposed method output for non-training samples?

---

### Official Review · Reviewer_PoRf · 2023-10-31

**Soundness:** 2 fair
**Presentation:** 3 good
**Contribution:** 2 fair
**Rating:** 5
**Confidence:** 2

**Summary:**

This paper focuses on developing methods that can recover degraded training data samples from overparameterized autoencoders (AE).

They formulate the task as an inverse problem and proposes an iterative optimization method to solve the optimization.

The proposed method significantly outperformed the baseline method on both FC AE & UNet AE and CIFAR-10 images.

**Strengths:**

The formulation of the recovery problem is interesting and makes sense to me.

**Weaknesses:**

My main concern is on the assumption and experimental setup.

This work assumes that the autoencoder can overfit to the real data. I'm unsure whether this could happen in practice if we train AE on a real-world dataset. The largest data the AE is trained on is 25,000 images. Wondering that will happen if we apply the proposed method on a AE trained on a  real-world data, e.g., the AE used in Stable DIffusion that has been widely used by tons of work on image generation.

The current experimental setup is limited to small-scale. Would be great to see results on large-scale setup.

**Questions:**

See Weakness

---

> ### Author Response · Authors · 2023-11-12
>
> Dear Reviewer, thank you for your time and comments.
>
> Indeed, the submitted paper includes various experiments where the maximal dataset size is 25,000 images from Tiny ImageNet. We can add experiments that consider dataset sizes of up to 100,000 images from Tiny ImageNet (i.e., the entire Tiny Image dataset). Please let us know if you would consider to raise your review score for such an addition to the paper.
>
> Regarding using autoencoders from the Stable Diffusion model: Please note that the recovery method in the submitted paper is intended for autoencoders that were trained using a squared error loss without explicit regularization. Specifically, the recovery method in the submitted paper is not intended for strongly regularized autoencoders (e.g., for denoising) and not for variational autoencoders. Due to this reason, we believe that we cannot perform your suggested experiment with the pretrained autoencoders from the Stable Diffusion.
>
> Based on the above, we believe that adding experiments that increase our maximal dataset size from 25,000 to 100,000 training samples is a fair way to address your concern. We would appreciate to understand if you would consider to significantly increase your review score for such addition to the paper.

---

### Official Review · Reviewer_RTXb · 2023-11-02

**Soundness:** 2 fair
**Presentation:** 3 good
**Contribution:** 2 fair
**Rating:** 3
**Confidence:** 4

**Summary:**

The consider the linear inverse problem of blind inpainting for a particular set of images which are used priory to train an overparameterized autoencoder. The use ADMM and the trained AE in a plug-and-play prior scheme to estimate the degradation operator on the training image, and fully recover the training image.

**Strengths:**

This problem is of particular interest to both the inverse problem community, and those interested in privacy issues concerning revealing the training data of a trained network. The paper is written clear and was easy to follow.

**Weaknesses:**

I find some motivations of the framework not well-suited for wider applications. They consider recovery of training images from degraded measurement of the image. How is this method applicable to inverse problems on data not appeared in the training data? How is the recovery when there is no measurement of the training image? The authors indeed shows (ans to my first question) that their approach cannot be used to solve general inverse problems on a test set. How applicable this method is on inverse problems with other measurement operators (e.g., additive Gaussian noise, Gaussian blurring, random inpainting, motion blur)? The experiments are not comprehensive.


The paper applies plug-and-play approach which is used is solving a general inverse problem using a denoiser on recovery of training images; they replace the denoiser with an autoencoder (the concept is exactly the same). The contribution of the method for inverse problem is limited.

Prior works on AE have shown that AE can recover training images. However, the reported numbers for the baseline is super low.

See my questions.

Minor comments

- I do not find the usefulness of the Theorem 1 for the general overparameterized, untied autoencoder. In general, (16) has been used and well-motivated by the plug-and-prior literature, so not clear why the authors on discussing usage of a network as a proximal operator for implicit regularization.

-  Please provide appropriate citations for alternating-minimization method (6), (7). This is a well-known procedure in dictionary learning. One example is [1].

- The last two paragraphs in Section 1 statements are very vague (has no citation) and not clear what the exact comparison is. See my question in Q section.


[1] Chatterji, N. S., & Bartlett, P. L. (2017). Alternating minimization for dictionary learning: Local convergence guarantees. arXiv preprint arXiv:1711.03634.

**Questions:**

1. The paper consider linear inverse problem of blind inpainting. I wonder how the performance of the framework is on other linear inverse problem (additive Gaussian noise, Gaussian blurring, random inpainting) or non-linear inverse problems (motion blur)?

2. Can the authors elaborate how much was the "small training dataset size" that was used in prior works?  The authors argued that their method can be applied on large training set images unlike prior work. The main experimental results include 600 images, and 50 images. Could the author elaborate on large training set?

3. Could the author elaborate which prior methods the outperform by citing (above section 2)? Does prior works try to recover training images given some degraded measurement or without having any measurement from it? Please elaborate, as this is crucial for fair comparison between the methods.

4. "our results also demonstrate the reduction in the recovery ability as the autoencoder is trained to a
higher train loss and less overfits its dataset. This, as well as our other results, are useful to understand the privacy risk of training data recovery in autoencoders." Is this a new finding? or I find this trivial.

5. The definitions of image regularizer and H regularizer in (5) are not rigorous and is not defined. Please elaborate on the wording "probable", and how the regularizers are implemented. Are they smooth? differentiable? ...

6. What is the motivation toward using ADMM as opposed to vanilla gradient on regularized objective? Why the splitting provides benefit in this case?

7. For (18), why not defining H only as a diagonal matrix in the first place? Then (18) is not needed.

8. Can authors elaborate on how they define "recovery"?

9. Can the author explain why the "AE iteration only does not work"? Providing a visualization on the iterations on AE to find its fixed point can be helpful.

10. Possible to visualize some failed examples?

---

> ### Author Response · Authors · 2023-11-12
>
> Dear Reviewer, thank you for your time spent on reviewing our paper.
> Our detailed response below (provided in two comments due to space limits) answers your questions, as well as suggests possible additions to the paper based on your requests. We would appreciate to understand if you would consider to significantly increase your review score based on our answers and the proposed updates of the paper.
>
>
> Response to Questions:
>
> 1.	We developed our method for a general degradation operator and tested it only for blind inpainting. Based on the related literature of training data recovery and of plug and play priors, we believe that blind inpainting is a sufficiently intricate and informative case, and that experiments for other degradation operators are not really necessary.
> Nevertheless, we are willing to add experiments for other degradation operators (e.g., from the list the reviewer provided) in non-blind settings (i.e., with a known degradation operator) if the reviewer would consider to significantly raise the review score. Dear reviewer, please let us know about it soon, so we will have enough time to add such experiments.
>
>
> 2.	Please note the last paragraph of Section 4 where we discuss our results for datasets of 1000 and 25,000 training samples. These results are shown in Tables 1,2 in page 9 of the submitted paper.
>
> 3.	The previous works use a simple iterative application of the autoencoder to recover training images from degradation of missing pixels, see for example the paper by Radhakrishnan et al. (2020) that we cite in our paper. Therefore, the setting that we have for blind inpainting is a completely fair comparison.
>
> 4.	Our new inverse problem perspective significantly improves the training data recovery performance in the examined settings. Given this new performance, we show that the recovery performance reduces as the autoencoder is trained to a higher train loss. We show this behavior in detail throughout our experiments and consider it as one of the contributions of our paper.
>
> 5.	Assumptions on the smoothness or differentiability of the implicit regularizer might be required for a theoretical analysis of the iterative optimization, but we anyway do not include such analysis due to its very weak relation to our blind inpainting setting. For the purpose of a practical application of plug and play priors, there is no need to explicitly demand that the implicit regularizer would be smooth nor differentiable. This is because the implicit regularizer is later replaced with a black box application of the autoencoder (see Equations (13) and (16)).
>
> 6.	The variable splitting and ADMM lead to the plug and play priors approach that lets us to apply an autoencoder as a black box without knowing the details of its implicit regularizer for the training data. Specifically, note that the optimizations in Equations (5) and (6) cannot be directly addressed (not by gradient descent and not otherwise), simply because the regularizer s_f is implicitly defined, and only later the optimization in Equation (13) can be replaced with the autoencoder application as a black box.
>
> 7.	The indicator function in Eq. (18) does not only requires H to be diagonal but also that its main diagonal components will be 0 or 1.
>
> 8.	The recovery definition is provided in Section 2.2 around Eq. (4) and in the second paragraph of Section 4 of the submitted paper.
>
> 9., 10., We can add such visualizations if the reviewer would consider to significantly raise the review score.

---

> > ### Author Response · Authors · 2023-11-12
> >
> > Re Weaknesses:
> >
> > •	Motivation: Our work is motivated by several previous works (cited in our paper) that examined the recovery of training data from overparameterized autoencoders. The motivation to these previous works was to better understand the associative memory in overparameterized autoencoders, and their recovery approach was simply to iteratively apply the autoencoder on a training image with missing pixels. These previous works have achieved nice recovery results, but only for limited settings such as particular activation functions in the neural networks and sufficiently small training datasets.
> > Accordingly, the main motivation for our work is to show that, by using an inverse problem perspective, the recovery of training data can be significantly improved and successfully applied on many more architectures, larger datasets, and higher train loss values (i.e., less overfitting) than in previous works. By that, we believe that our work significantly contributes to the understanding of memorization in autoencoders and to the understanding of the potential ability of training data recovery from trained models.
> >
> > •	Applying our recovery method on data not from the training dataset:
> > Our method is designed to recover training data of a given autoencoder. It is based on the assumption that a trained overparameterized autoencoder includes an implicit prior (regularizer) for its specific training dataset. Accordingly, there is no expectation that our recovery method will work well on data not from the specific training dataset; furthermore, we expect that our method will not work well on such non-training data.
> > To confirm this, our submitted paper already includes an experiment where our method is applied for a blind inpainting of degraded images that were not included in the training dataset (but, importantly, these images are from the same overall dataset and therefore have similar general characteristics as the training images). Indeed, we reported in the submitted paper (see the fifth paragraph in Section 4) that our method cannot recover test (i.e., non-training) data, which is the reasonable and expected result here.
> >
> > •	Other degradation (“measurement”) operators: See our above answer to your first question.
> >
> > •	Reported performance of baseline recovery method: As we mentioned in the paper, and also above here, please note that previous works indeed showed good recovery performance but only for a limited number of architectures (e.g., for particular activation functions) and for sufficiently small training datasets. The reported performance for the baseline recovery is relatively low in our experiments simply because we considered activation functions and training dataset sizes that are out of the scope where the baseline method is known to work well. Therefore, there is no problem with the baseline evaluation. In fact, our results clearly show the significance and great performance gains of our inverse problem approach.
> >
> > •	Relation to the plug and play priors: Please note that the original plug and play priors method suggests to employ an image denoiser, whereas we suggest to employ an autoencoder that was trained to output its input and not to denoise. We believe that this is a delicate difference that should be described in our paper, as we indeed do.
> >
> > •	Additional references: We can cite the references the reviewer requested for alternating minimization and for the last two paragraphs in Section 1.

---

> > ### Comment · Reviewer_RTXb · 2023-11-16
> > **The approach cannot solve inverse problems on test set.**
> >
> > I thank the authors for their response. My main concern is that this approach cannot be used to solve general inverse problems on a test set (which is shown by the author).
> >
> > 1. In addition to masking, where certain pixels are left unchanged, I recommend adding other forms of inverse problems such as denoising, Gaussian blur. It's not clear if the method and network would be able to recover successfully when all pixels are corrupt.

---

### Author Response · Authors · 2023-11-16

Dear AC and Reviewers,

Thank you for your time spent on reviewing our paper. We would like to update that we have decided to withdraw our paper from this review process. It is important to emphasize that we are confident in the quality and contributions of our paper.

In addition to the two replies that we posted here below earlier this week, we also would like to emphasize that the submitted paper already included some of the details and experiments that were requested by some of the reviewers. Specifically, please note that the evaluation of the proposed recovery method for non-training data is discussed in the fifth paragraph of Section 4 in the submitted paper.

Thanks again for your time.